# Origin of conformational dynamics in a globular protein

Adam M. Damry[1], Marc M. Mayer[1], Aron Broom[1], Natalie K. Goto[1] & Roberto A. Chica [1]*

Protein structures are dynamic, undergoing motions that can play a vital role in function. However, the link between primary sequence and conformational dynamics remains poorly understood. Here, we studied how conformational dynamics can arise in a globular protein by evaluating the impact of individual core-residue substitutions in DANCER-3, a streptococcal protein G domain β1 variant that we previously designed to undergo a specific mode of conformational exchange that has never been observed in the wild-type protein. Using a combination of solution NMR experiments and molecular dynamics simulations, we demonstrate that only two mutations are necessary to create this conformational exchange, and that these mutations work synergistically, with one destabilizing the native structure and the other allowing two new conformational states to be accessed on the energy landscape. Overall, our results show how dynamics can appear in a stable globular fold, a critical step in the molecular evolution of dynamics-linked functions.

[1] Department of Chemistry and Biomolecular Sciences, University of Ottawa, 10 Marie-Curie, Ottawa, ON, Canada K1N 6N5. *email: rchica@uottawa.ca

Proteins carry out most of the intricate biological processes required to sustain life. These functions often involve motions[1–3], with conformational dynamics playing a vital role in enzyme catalysis[4,5], allosteric regulation[6], and molecular recognition[7]. Despite the demonstrated importance of dynamics for protein function, the link between protein sequence and dynamics remains poorly understood[8], and epistasis resulting from the ruggedness of the protein energy landscape complicates attempts to study how sequence elements contribute to dynamics and thereby function in natural proteins[9]. Progress towards this goal has been made by evolution-based studies[10–12], with key findings having shown that novel protein functions can arise from new dynamic regimes that reorganize functional sites[13,14]. However, these studies have focused on function-expanding dynamics that develop in metastable regions of proteins where preexisting motions responsible for a promiscuous activity can be amplified and modulated with small changes to protein sequence and structure[10,15]. Therefore, a fundamental understanding of how sequence changes can lead to novel modes of conformational exchange in globular proteins remains elusive.

Previously, we used computational design to create a series of three streptococcal protein G domain β1 (Gβ1) variants termed DANCERs (DANCER-1, DANCER-2, and DANCER-3)[16] that undergo a specific mode of conformational exchange that has never been observed in the wild-type protein[17–26]. This exchange involves spontaneous switching of the Trp43 side chain between core-buried and solvent-exposed states[16], which is accompanied by rotameric changes to residues in the protein core and by displacement of the helix relative to the β-sheet that nevertheless preserves the native Gβ1 fold (Fig. 1a). The successful generation of DANCERs involved simultaneous introduction of five or six mutations to the Gβ1 core that were predicted to stabilize both target states and create a sufficiently small transition state barrier to allow interconversion on a functional timescale. In contrast with other proteins previously used to study the evolution of dynamics[10,15,27–30], DANCERs are unique in that the designed Trp43 conformational exchange did not result from alteration of exchange rates and amplitudes between existing low-occupancy conformational states observed in the parent. Wild-type Gβ1 and DANCERs therefore comprise an ideal system to study how conformational exchange can arise in a globular protein, helping us to gain insights on the link between protein sequence and dynamics.

Here, we evaluated the impact on Trp43 conformational exchange of individual core-residue substitutions in a representative DANCER in order to understand the role of each residue in the designed dynamic exchange. Solution NMR experiments show that only two mutations were necessary to

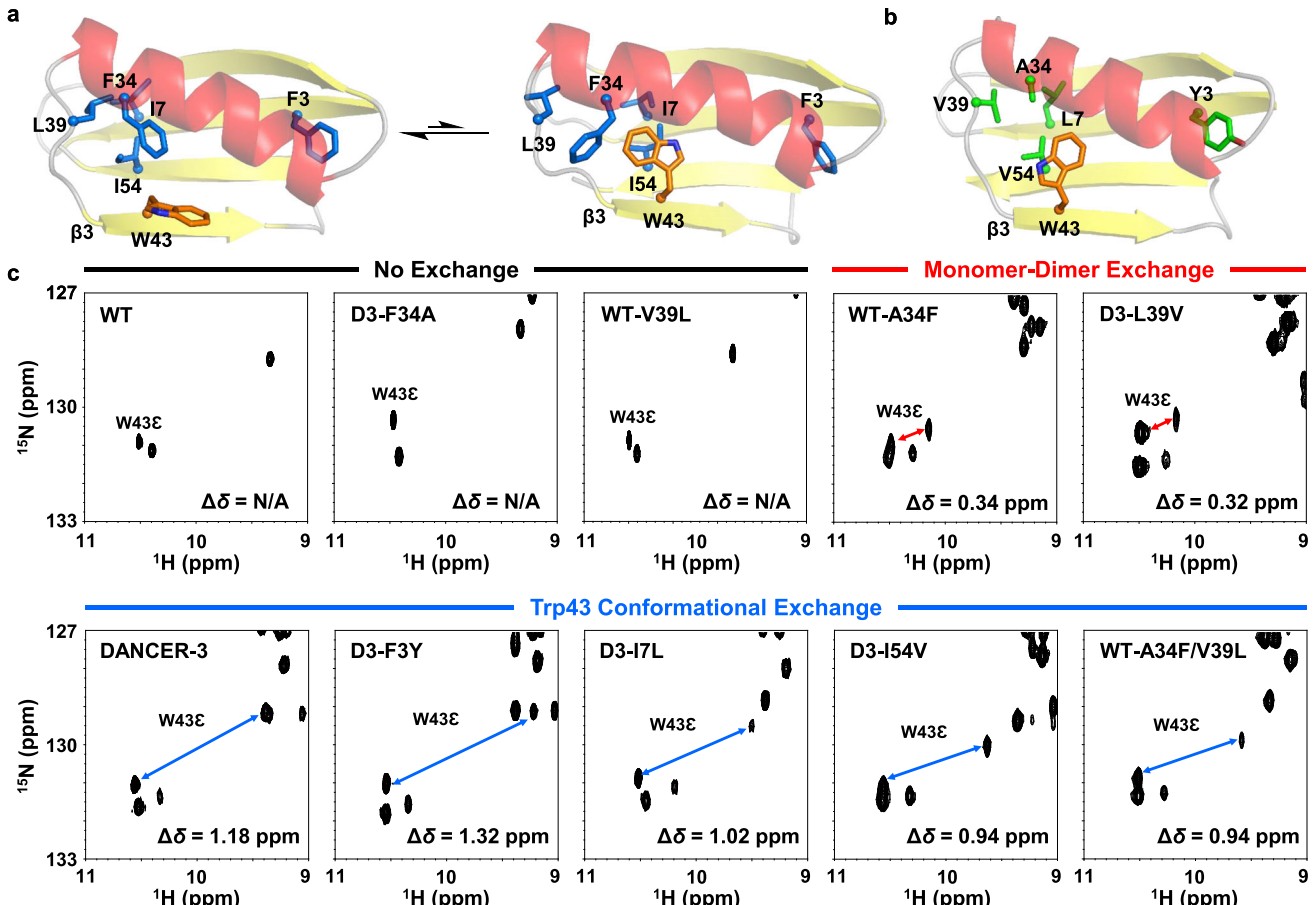

**Fig. 1** Contribution of individual mutations to exchange. **a** Structures of the DANCER-3 major and minor states (modeled on PDB ID: 5UBS and 5UCF, respectively) and **b** Wild-type Gβ1 (PDB ID: 1PGA) show the side chains of W43 (orange) and core residues at mutated positions (blue or green for DANCER-3 or wild-type Gβ1, respectively). α-helices, β-strands, and loops are colored red, yellow, and gray, respectively, and strand β3 is labeled at its N-terminus. **c** $^1$H-$^{15}$N HSQC spectra of selected Gβ1 variants. W43ε minor state peaks identified using ZZ-exchange experiments are shown linked to their corresponding major state peak with a double-sided arrow. Chemical-shift displacements (Δδ) between major and minor state peaks are indicated (N/A = not applicable). Variants were grouped as displaying no exchange, monomer-dimer exchange, or Trp43 conformational exchange, according to the magnitude of their calculated Δδ.

convert the static wild-type Gβ1 into a DANCER capable of undergoing Trp43 conformational exchange. The molecular mechanism by which these mutations induce Trp43 conformational exchange was probed using molecular dynamics simulations, showing that these mutations work synergistically, with one destabilizing the native Gβ1 structure and the other allowing two new conformational states to be accessed on the energy landscape. Our results thus provide a model for the molecular evolution of conformational dynamics in globular proteins.

## Results

**Contribution of individual mutations to exchange.** To investigate how sequence changes can lead to novel modes of conformational exchange in a globular protein, we individually reverted each of the five mutations present in DANCER-3 to the corresponding amino acid found in wild-type Gβ1 (Fig. 1a, b). We chose DANCER-3 for this analysis over other DANCERs as its slow exchange rate and distinct $^1$H and $^{15}$N chemical shifts for both Trp43 conformational states facilitate analysis of exchange using nuclear magnetic resonance (NMR) spectroscopy. The resulting five DANCER-3 point mutants (D3-F3Y, D3-I7L, D3-F34A, D3-L39V, and D3-I54V, each named for the mutation they possess relative to DANCER-3) all adopted the native Gβ1 fold (Supplementary Fig. 1) and were stably folded (Table 1, Supplementary Figs. 2 and 3). $^1$H-$^{15}$N heteronuclear single quantum coherence (HSQC) spectra for all variants with the exception of D3-F34A showed a population of additional peaks not observed in the wild-type Gβ1 spectrum (Supplementary Figs. 4 and 5), indicating that these variants adopt more than one distinct structural state, as observed for DANCER-3[16]. Given the difference in NMR spectra between D3-F34A and all other DANCER mutants studied here, we investigated the possibility that DANCER variants containing a Phe residue at position 34 (Table 1) form dimers in solution, as the A34F mutation has been shown to induce dimerization when introduced on its own into wild-type Gβ1[25,31]. Analytical size-exclusion chromatography using wild-type Gβ1 and its A34F point mutant (WT-A34F) as controls for monomeric and dimeric proteins, respectively, revealed that unlike wild-type Gβ1, which exists as a monomer under these conditions, DANCER-3 and all its variants containing the A34F mutation form weak dimers with $K_d$ values measured by NMR between 31 and 240 μM (Supplementary Fig. 6 and Supplementary Table 1). Although monomer-dimer equilibrium could give rise to the observed populations of peaks in HSQC spectra, we previously confirmed that the Trp43 side chain of DANCERs undergoes conformational exchange by demonstrating that the nuclear Overhauser effect (NOE) contacts made by this residue

cannot arise from its adoption of a single conformation in the context of the native Gβ1 fold, and by solving solution structures of the major and minor states of the DANCER-2 variant[16]. As the Trp43 conformation in the WT-A34F dimer structure is identical to that of the wild-type Gβ1 monomer (Fig. 1b and Supplementary Fig. 7), these contacts cannot be explained by monomer-dimer equilibrium. Furthermore, the dimer interface lies on the opposite face of the protein from Trp43 (Supplementary Fig. 7), which would only give rise to small changes in local chemical environment around Trp43 upon dimerization, whereas conformational exchange between buried and solvent-exposed states would be expected to give rise to larger changes.

To distinguish whether the second population of Trp43 peaks in our DANCER variants resulted from monomer-dimer equilibrium or Trp43 conformational exchange, we measured Trp43ε chemical shift displacements (Δδ) for these variants as well as WT-A34F and DANCER-3. We observed a smaller Trp43ε Δδ in the WT-A34F spectrum than that of DANCER-3 (0.34 ppm vs. 1.18 ppm, Fig. 1c). HSQC spectra of D3-F3Y, D3-I7L, and D3-I54V showed a Δδ close to that of DANCER-3 (0.94–1.32 ppm). Conversely, the spectrum of D3-L39V instead exhibited a Δδ similar to the one observed for WT-A34F (0.32 ppm), suggesting that D3-L39V does not undergo the Trp43 conformational exchange characteristic of DANCERs, with its Trp43ε Δδ arising from the monomer-dimer equilibrium as in WT-A34F.

To determine if these two peak populations interconvert on a millisecond timescale, as was observed for the Trp43 conformational exchange in DANCER-3, $^1$H-$^{15}$N HSQC ZZ-exchange experiments were performed on all variants (Supplementary Fig. 8). Although dynamics may differ between variants at multiple timescales, we focused on those at the millisecond timescale, since these were the target of our original design. We observed exchange for Trp43ε as well as Thr17/Thr18, which can be used as Gβ1 dimerization reporters due to their localization at the dimer interface (Supplementary Fig. 7). To determine whether Trp43ε exchange arises from monomer-dimer equilibrium, we compared activation energies for Trp43ε exchange with that of Thr17/Thr18. In both WT-A34F and D3-L39V, activation energies calculated from both Thr17 and Trp43ε were the same within error (Table 2, Supplementary Fig. 9), strongly suggesting that dynamics probed at both positions are representative of monomer-dimer exchange. In contrast, DANCER-3 rates for Trp 43ε minor to major state exchange decreased as a function of temperature (Fig. 2a), while Thr17 instead demonstrated the expected increase of exchange rate with temperature. This non-Arrhenius behavior indicates a complex exchange trajectory that

**Table 1 Stability of Gβ1 variants.**

| Protein | Mutations from wild type | | | | | $Tm^a$ (° C) | $Cm^b$ (M) | $\Delta G_U{}^c$ (kcal mol$^{-1}$) |
|---|---|---|---|---|---|---|---|---|
| | Y3 | L7 | A34 | V39 | V54 | | | |
| Wild type | – | – | – | – | – | 81 | 2.66 ± 0.04 | 4.3 ± 0.3 |
| WT-A34F | – | – | F | – | – | 57 | 1.02 ± 0.04 | 2.7 ± 0.2 |
| WT-V39L | – | – | – | L | – | 74 | 2.32 ± 0.02 | 4.3 ± 0.4 |
| WT-A34F/V39L | – | – | F | L | – | 61 | 1.34 ± 0.01 | 3.0 ± 0.1 |
| DANCER-3 | F | I | F | L | I | 71 | 1.72 ± 0.05 | 3.4 ± 0.4 |
| D3-F3Y | – | I | F | L | I | 66 | 1.66 ± 0.06 | 3.3 ± 0.3 |
| D3-I7L | F | – | F | L | I | 60 | 1.30 ± 0.03 | 2.8 ± 0.4 |
| D3-F34A | F | I | – | L | I | 84 | 2.74 ± 0.07 | 4.3 ± 0.1 |
| D3-L39V | F | I | F | – | I | 68 | 1.75 ± 0.05 | 2.9 ± 0.1 |
| D3-I54V | F | I | F | L | – | 65 | 1.82 ± 0.06 | 3.2 ± 0.4 |

$^a$Thermal denaturation midpoint temperature determined through loss of circular dichroism signal at 208 nm
$^b$Concentration of guanidinium chloride at denaturation midpoint (25 °C, $n = 3$ technical replicates, mean ± s.d.)
$^c$Free energy of unfolding determined by chemical denaturation with guanidinium chloride at 25 °C ($n = 3$ technical replicates, mean ± s.d.)

**Table 2 Monomer-dimer and Trp43 conformational exchange activation energies.**

| Sequence | Peaks | $E_a^{app}{}_{1\rightarrow2}$[a] (kcal/mol) | $E_a^{app}{}_{2\rightarrow1}$[b] (kcal/mol) | Arrhenius behavior[c] |
|---|---|---|---|---|
| WT-A34F | W43ε | 8 ± 1 | 10 ± 2 | Yes |
|  | T17 | 10 ± 1 | 12 ± 1 | Yes |
| D3-L39V | W43ε | 19 ± 2 | 16 ± 2 | Yes |
|  | T17 | 18 ± 2 | 17 ± 2 | Yes |
| DANCER-3 | W43ε | 9 ± 1 | −27 ± 3 | No |
|  | T17 | 7 ± 1 | 12 ± 1 | Yes |
| D3-F3Y | W43ε | 3 ± 1 | 1.6 ± 0.3 | Yes[d] |
|  | T17 | 13 ± 1 | 16 ± 2 | Yes |
| D3-I7L | W43ε[e] | N/A | N/A | N/A |
|  | T18 | 31 ± 3 | 25 ± 2 | Yes |
| D3-I54V | W43ε | 0[f] | 0[f] | No[d] |
|  | T17 | 8 ± 2 | 12 ± 1 | Yes |
| WT-A34F/ V39L | W43ε | 1.6 ± 0.2 | −4.7 ± 0.6 | No |
|  | T18 | 9 ± 1 | 13 ± 1 | Yes |

[a]Apparent activation energy for exchange from the major state (the most populated state at the lowest temperature tested) to the minor state (the least populated state at the lowest temperature tested) ($n = 2$ analytical replicates, mean ± s.d.)
[b]Apparent activation energy for exchange from the minor state to the major state ($n = 2$ analytical replicates, mean ± s.d.)
[c]Arrhenius behavior is defined here as a positive activation energy for both transitions
[d]Although D3-F3Y activation energies are both positive, they are weak compared to T17 (Supplementary Fig. 10). It is expected that for one process, they would be similar. This is not the case, thus different dynamic processes are being sensed by D3-F3Y Trp43ε and T17
[e]W43ε minor state peaks and crosspeaks could not be quantified for D3-I7L but are present
[f]Exchange rates did not vary as a function of temperature over the range measured

may be explained by competing effects between increasing rates of Trp43 side-chain dynamics and decreasing populations of monomeric Gβ1 at the higher temperatures (Fig. 2b, c). Therefore, the net reduction in measured Trp43ε exchange rates at higher temperatures suggests that this process is inhibited in the dimer. Our prior results with the L39I mutant of DANCER-3 (previously reported as NERD-S, here renamed D3-L39I)[16] corroborate this hypothesis, as D3-L39I exhibits no detectable Trp43 conformational exchange at the concentration range used in our NMR experiments (0.1–1 mM) despite high sequence and spectral similarity to DANCER-3[16]. At a concentration of 100 μM, the monomeric form of DANCER-3 was observable while that of D3-L39I was not. However, when the concentration of D3-L39I was lowered to 20 μM, which is close to its $K_d$ (Supplementary Table 1), we observed the appearance of a second Trp43ε peak with a Δδ characteristic of Trp43 conformational exchange (Supplementary Fig. 10). Non-Arrhenius behavior for Trp43ε similar to that observed for DANCER-3 was also observed in D3-F3Y, D3-I7L, and D3-I54V (Table 2, Supplementary Fig. 11), confirming that these three mutants exhibit Trp43 conformational exchange similar to that of DANCER-3. In contrast, D3-F34A, which exhibits only a single population of peaks in its HSQC, does not undergo Trp43 conformational or monomer-dimer exchange, and adopts a structure highly similar to that of wild-type Gβ1 (Supplementary Fig. 12, Table 3). Altogether, these results confirm that DANCER-3 mutants D3-F3Y, D3-I7L, and D3-I54V undergo Trp43 conformational exchange, but that this mode of molecular motion does not occur in wild-type Gβ1, WT-A34F, D3-F34A, or D3-L39V.

**F34 and L39 synergistically create conformational exchange**. Having identified Phe34 and Leu39 as two key residues required for Trp43 conformational exchange in DANCER-3, we next investigated whether these mutations could create this mode of conformational exchange when introduced together into wild-type Gβ1. The resulting WT-A34F/V39L double mutant adopted the native Gβ1 fold, was stable, and formed a dimer with a $K_d$ of

116 ± 5 μM (Table 1, Supplementary Figs. 1–3 and 6, Supplementary Table 1). Its HSQC spectrum demonstrated the presence of two peak populations (Supplementary Fig. 5) with a Δδ of 0.94 ppm (Fig. 1c) for the Trp43 side-chain peak. Furthermore, temperature-dependent exchange of the minor state back to the major state for W43ε, as monitored by ZZ-exchange spectra, showed non-Arrhenius behavior (Supplementary Figs. 8, 9, 11), indicating that Trp43 conformational exchange does occur when both mutations are introduced simultaneously. In contrast, the WT-V39L point mutant, which adopted the native Gβ1 fold and was stable (Table 1, Supplementary Figs. 1–3), was instead monomeric and gave rise to an HSQC spectrum with high similarity to that of wild-type Gβ1 (Supplementary Figs. 6 and 13), demonstrating that its Trp43 conformation is the same as in wild-type Gβ1. Taken together, these results show that neither V39L nor A34F alone is sufficient to alter the conformation of Trp43 or create dynamic exchange of this residue when introduced into wild-type Gβ1. There is therefore a synergistic effect between Leu39 and Phe34 that leads to the specific mode of Trp43 conformational exchange that we designed into DANCERs.

**L39 stabilizes loop connecting helix with strand β3**. To gain insights on the role of the conservative V39L mutation in Trp43 conformational exchange and evaluate its underlying synergistic effect with A34F, we performed microsecond-timescale molecular dynamics simulations on DANCER-3, D3-F34A, and D3-L39V. We also included D3-L39I in this analysis because it contains an isosteric mutation at position 39 that might further help to demonstrate the role of this residue in DANCER dynamics. Although the timescale of these simulations is shorter than that of the ZZ-exchange experiments, motions observed at the microsecond timescale in molecular dynamics may form part of the millisecond timescale processes revealed by NMR. Over the course of each 10-μs trajectory, the Trp43 side chain in these proteins samples a number of different conformations, defined by their $\chi_1$ and $\chi_2$ dihedrals (Supplementary Fig. 14). The number of transitions between these Trp43 conformations over the course of the simulation is highest in DANCER-3, and close to zero in D3-F34A (Fig. 3a), in agreement with the observed presence or absence of Trp43 conformational exchange measured by NMR (Fig. 1c). D3-L39V also undergoes very few transitions during the course of the molecular dynamics simulations, which is consistent with the fact that Trp43 conformational exchange in this variant was not detected by NMR (Fig. 1c, Table 2). The D3-L39I monomer undergoes the second highest number of transitions even though no measurable Trp43 conformational exchange was observed at the concentrations used for NMR at which D3-L39I is predominantly in the dimeric form. This result, along with a lack of NOEs observed from the W43 H$_{\epsilon 1}$ to the protein core[16], supports our hypothesis that only the monomeric form of DANCERs is capable of undergoing Trp43 conformational exchange.

Molecular dynamics also allowed us to observe other motions in close proximity to Trp43 that may be coupled with its conformational exchange (defined here as a transition between $g(+)$, $g(-)$, or $t$ conformations for the $\chi_1$ dihedral). These simulations revealed one mode of Phe34 motion that appeared to be correlated with Trp43 conformational exchange in DANCER-3 and D3-L39I (Fig. 3b, Supplementary Fig. 14c). Specifically, the introduction of a large aromatic side chain at this position stabilizes a state where the Trp43 side chain has rotated out of the protein core by adopting the $g(+)$ conformation, since the Phe34 side-chain can occupy the cavity that would otherwise be created by the Trp43 rotation[16]. In D3-F34A, the smaller Ala34 side

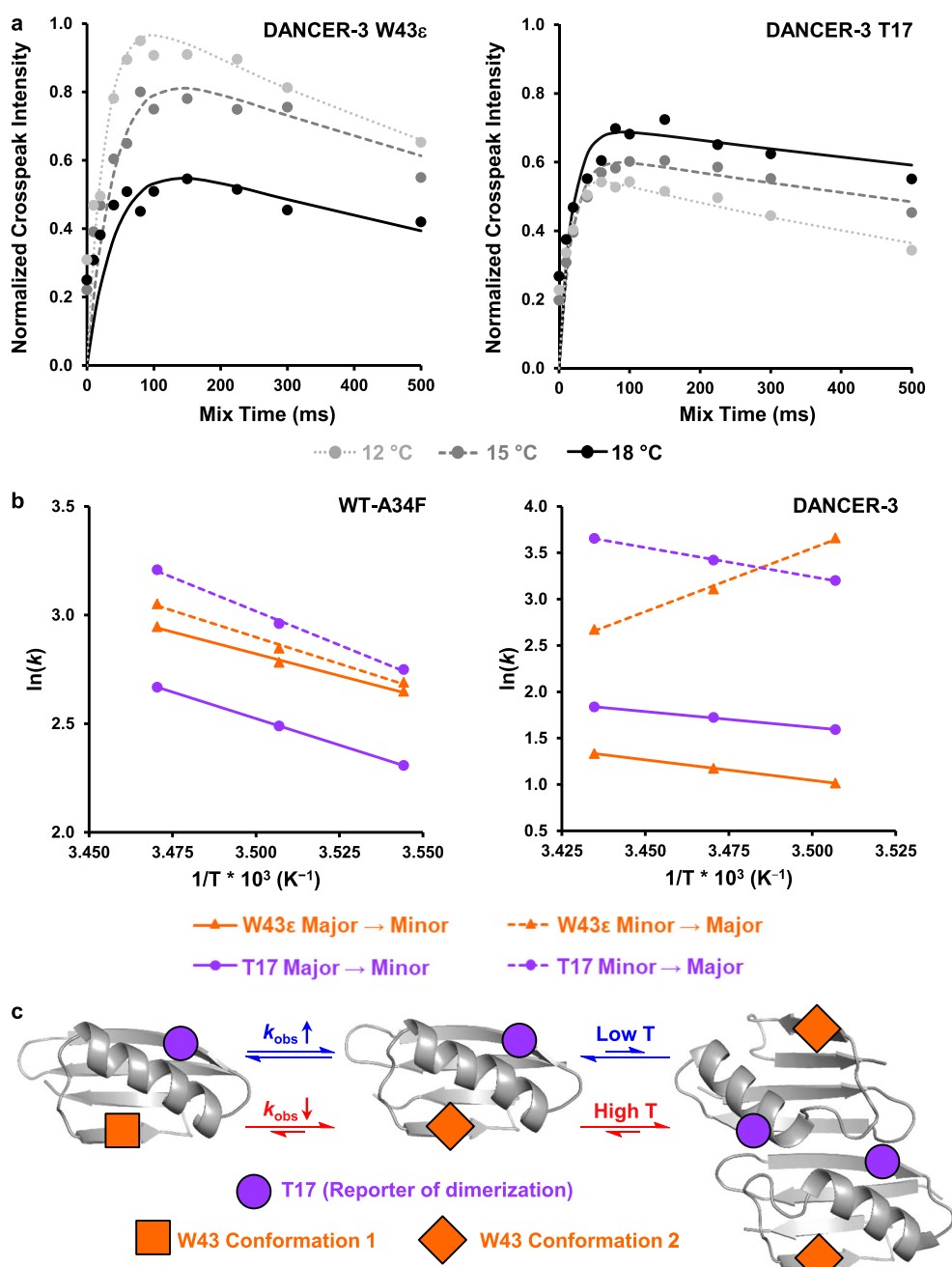

**Fig. 2** Trp43 conformational exchange does not follow Arrhenius behavior in DANCERs. **a** $^1$H-$^{15}$N HSQC ZZ-exchange crosspeak intensity curves for the W43ε minor to major state transition show that this transition slows in DANCER-3 as temperature increases. This unusual behavior is not observed with T17, demonstrating a different mode of exchange for W43ε than monomer-dimer exchange. **b** Arrhenius plots demonstrate Arrhenius behavior for both W43ε and T17 exchange profiles in WT-A34F, as demonstrated by similar slopes (i.e., activation energies) for all four transitions. DANCER-3 however exhibits non-Arrhenius behavior, as shown by a W43ε minor to major state transition with a non-physical negative activation energy, which indicates that this residue does not follow a two-state Arrhenius model. In contrast, a normal Arrhenius behavior is observed for DANCER-3 T17. **c** In DANCERs, non-Arrhenius behavior for W43ε transitions is observed due to an altered monomer-dimer equilibrium. As dimerization of DANCERs is favored at high temperature and W43 exchange is inhibited in the dimer, observed rates ($k_{obs}$) of W43 exchange are reduced when temperature increases. In contrast, $k_{obs}$ for exchange of the T17 residue found at the dimer interface increase with temperature, indicating that dimerization kinetics follow Arrhenius behavior. Non-Arrhenius behavior for W43ε transitions was not detected in Gβ1 variants that do not undergo W43 conformational exchange, suggesting that all kinetics measured for these variants report only on dimerization.

chain cannot serve this purpose, and therefore the state with Trp43 in the $g(+)$ conformation is not energetically accessible, which prevents this conformational exchange. The role of Leu39 is less clear, as it exhibits no concerted transitions with Trp43. However, we observed that D3-L39V displays enhanced motion in the loop connecting the helix with strand β3, which contains

position 39 (Fig. 1a, b), compared to DANCER-3, D3-F34A, and D3-L39I (Fig. 3c). Moreover, this loop in D3-L39V frequently occupies an alternate conformation created by partial unfolding of the helix at its C-terminus (Fig. 3d), which positions residue 39 away from Phe34 and thereby creates a state that was seldom observed in simulations run for the other variants. A similar loss

**Table 3 Summary of NOE restraints and structural statistics.**

| PDB ID | 6NJF |
|---|---|
| NMR distance and dihedral restraints | |
| Distance constraints | |
|  Total NOE | 618 |
|  Short range ($|i - j| \leq 1$) | 336 |
|  Medium range ($1 < |i - j| < 5$) | 88 |
|  Long range ($|i - j| \geq 5$) | 194 |
| Total dihedral angle restraints | |
|  $\varphi$ | 64 |
|  $\psi$ | 64 |
| Structure statistics | |
| Violations | |
|  Distance constraints (>5 Å) | 0 |
|  Dihedral constraints (>0.5°) | 0 |
| MolProbity Ramachandran plot statistics (%) | |
|  Residues in most favored regions | 96.3 |
|  Residues in allowed regions | 3.7 |
|  Residues in disallowed regions | 0.0 |
| Average pairwise RMS deviation (Å) | |
|  Backbone (mean ± 1S.D.) | 0.29 ± 0.06 |
|  Heavy Atom (mean ± 1S.D.) | 0.79 ± 0.09 |
| Structure quality factors (Raw/Z-score[a]) | |
|  MolProbity clash score | 16.85/–1.37 |
|  Procheck G-factor (phi & psi) | –0.26/–0.71 |
|  Procheck G-factor (all) | –0.41/–2.42 |
|  Verify 3D | 0.42/–0.64 |
|  Prosall (negative) | 0.56/–0.37 |

Analyzed for the 10 lowest energy structures for each designed protein using CYANA[42] and MolProbity[43]
[a]With respect to mean and standard deviation for a set of 252 X-ray structures with sequence lengths ≤ 500, resolution ≤ 1.80 Å, and R-free ≤ 0.28

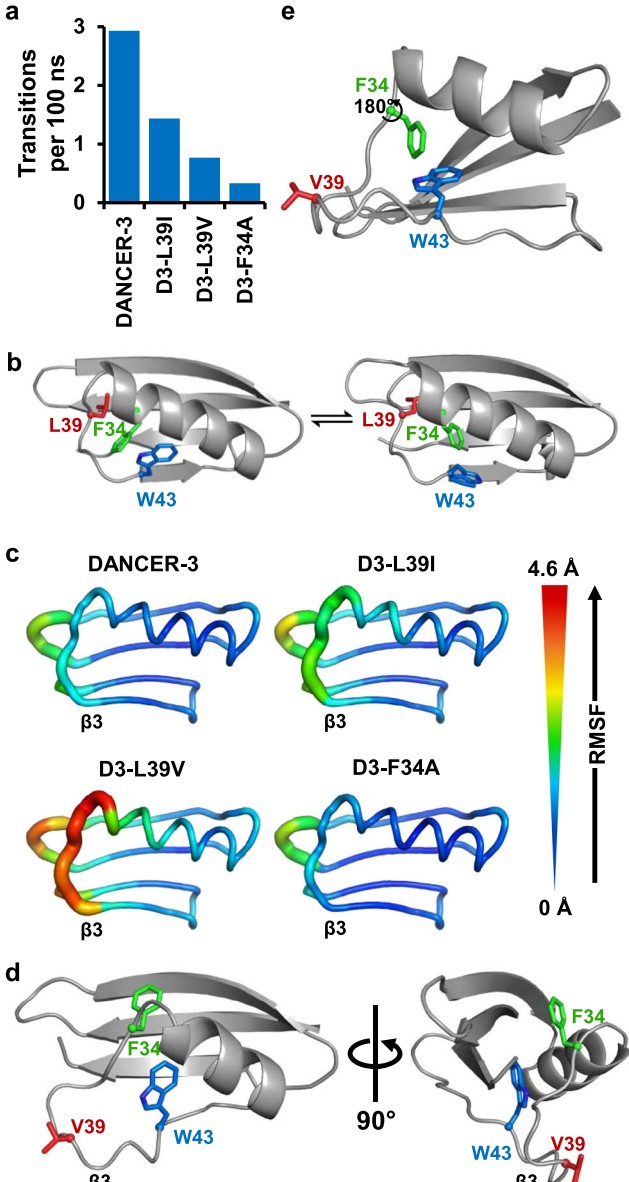

**Fig. 3 Molecular dynamics results. a** The number of Trp43 conformational transitions, defined as $\chi_1$ and $\chi_2$ dihedral changes between bins centered at −60°, 180°, and +60°, or −90° and +90°, respectively, decreases as a function of experimentally-validated dynamicity. **b** Representative snapshots sampled from the DANCER-3 simulation demonstrating the concerted motion between Phe34 and Trp43. **c** Backbone root mean square fluctuation (RMSF) over each simulation plotted on a model backbone for each respective protein. Increased RMSF seen in D3-L39V at the C-terminus of the α-helix and in the loop containing position 39, which connects the α-helix to strand β3 (labeled at its N-terminal extremity), results from increased mobility and ability to adopt alternate backbone conformations. **d** D3-L39V populates an alternate loop conformation comprising a partially unfolded α-helix. **e** Phe34 is incapable of displacing Trp43 while satisfying the backbone conformation described in panel **d** as shown by a model demonstrating that a 180° $\chi_1$ rotation of the Phe34 side-chain could be allowed without substantial displacement of the Trp43 side-chain.

of helicity in this region of the protein has also been seen in the WT-A34F solution structure (Supplementary Fig. 7). The increase in conformational flexibility in this region may decouple Phe34 side-chain reorientation from that of Trp43 (Supplementary Fig. 14c), since conformational dynamics in the backbone could accommodate the different Phe34 orientations in a way that prevents steric clashes with the native Trp43 side-chain conformation (Fig. 3e). On the other hand, in DANCER-3 the loop containing Leu39 predominantly adopts the native conformation, positioning Leu39 adjacent to Phe34 and maintaining a helical state in the backbone of residue 34 (Supplementary Fig. 14). This more stable backbone conformation is required in order for the Phe34 side chain to move in concert with that of Trp43, enabling Trp43 conformational exchange.

## Discussion

Our results demonstrate that two mutations synergistically create a novel mode of conformational exchange in DANCERs. The first mutation, A34F, positions a bulky side chain directly next to Trp43, likely leading to steric clashes that destabilize the protein by 1.6 kcal/mol (Table 1). These clashes could be alleviated by movement of the Trp43 side chain outside of the core. However, when A34F is introduced into wild-type Gβ1, it is instead the helix and the loop connecting it to strand β3 that undergo a conformational change, which minimizes steric clashes between Phe34 and Trp43. Meanwhile, the V39L mutation does not significantly impact the free energy of unfolding when introduced into wild-type Gβ1. In the context of DANCERs, however, this mutation prevents local unfolding of the Gβ1 α-helix, shifting the equilibrium of the loop towards the native conformation. This in turn forces Trp43 to move out of the core, leading to conformational exchange due to the equilibrium between the solvent-exposed and core-buried Trp43 side-chain conformations. Interestingly, the

addition of a single methylene group to residue 39 is sufficient to prevent local unfolding of the helix, demonstrating that a subtle structural change to a key region of a protein can give rise to a large effect on dynamics. Yet, it is not sufficient to simply overpack the core to create dynamics, since mutations such as V54I do not lead

to Trp43 conformational exchange in Gβ1, and DANCERs are at least as stable as WT-A34F, which has not been shown to undergo this mode of exchange. Instead, it is the introduction of a specific combination of mutations that remodels the energy landscape in a way that favors the emergence of dynamics.

Epistasis caused by ruggedness in the protein energy landscape complicates analysis of the evolution of new modes of conformational exchange in natural proteins. However, the Gβ1 variants described here provide a unique opportunity to start addressing this question. Although the conformational exchange of Trp43 observed in DANCERs was designed computationally, we showed herein that only two mutations are necessary for this dynamic exchange to arise. There are thus only two possible evolutionary pathways by which these mutations might be introduced (Fig. 4). As protein stability is an important determinant of evolutionary fitness[32], if both mutations were highly destabilizing, DANCERs would be evolutionarily inaccessible. However, the V39L mutation has no substantial effect on protein structure, stability, and function in the context of the wild-type Gβ1 sequence. Thus, it strongly resembles the "permissive mutations" observed during protein evolution[33], which have no immediate structural or functional effect but stabilize specific structural elements such that the protein can tolerate ensuing mutations causing a functional change. In the context of DANCERs, we postulate that V39L is a permissive mutation that reshapes the protein's energy landscape to disfavor the alternate loop conformation by stabilizing the helix at its C-terminus and thereby favor Trp43 conformational exchange without substantially altering the energy well depth of the native fold (Fig. 4). Conversely, the A34F mutation is highly destabilizing to the native fold and promotes the alternate loop conformation, facilitating access to other energy wells in the surrounding landscape. This explains the synergistic effects of these two mutations, where one destabilizes the native energy well and the other makes neighboring wells more accessible in a way that allows novel dynamic motions to develop.

Based on these results, we propose a hypothetical model for the molecular evolution of conformational dynamics in globular proteins. First, a permissive mutation such as V39L that silently reshapes unexplored regions of the protein energy landscape could spontaneously emerge. This mutation would not result in any phenotypic change to the protein but would remodel the energy landscape in a way that would increase accessibility to alternate states, for example by lowering barriers between these states, and stabilize them such that they could be substantially populated. Next, a destabilizing mutation such as A34F could appear and cause the emergence of a new dynamic process by making the native state less stable than the alternate conformations now made accessible by the permissive mutation, allowing these states to be substantially populated and interconvert. The combination of these two mutations would then form the basis for the evolution of a new function, as it has been shown that plasticity enhances evolvability[34]. Once these function-expanding mutations have been introduced, additional mutations could afterwards emerge to recover stability, such as the stabilizing Y3F, L7I, and V54I mutations in DANCER-3, or, with the right selective pressure, to fine-tune this new dynamic regime in order to create a novel function such as allosteric regulation or molecular recognition. This process parallels previously studied evolutionary processes wherein the development of new dynamics-linked function occurs via the introduction of mutations that amplify dynamics, followed by selection against undesirable competing motions that negatively impact activity or stability[10]. Previous studies have demonstrated how a small number of mutations can abolish and re-establish conformational exchange pathways in larger natural proteins[3,35], as we observed with our model system comprised of a relatively small

protein. These trends support the general idea that permissive mutations in globular proteins can provide a route for the introduction of new dynamic modes in the molecular evolution of dynamics-linked functions, shedding light on the link between protein sequence and dynamics.

## Methods

**Mutagenesis.** Codon-optimized and His-tagged (N-terminus) genes for wild-type Gβ1 (GenBank entry M13825.1) and DANCER-3 cloned into the pJ414 vector were obtained from ATUM (formerly DNA2.0). All other variants tested here were generated through splicing by overlap extension mutagenesis[36] using either wild-type Gβ1 or DANCER-3 as starting templates, and cloned into the pET-11a vector (Novagen) via the NdeI/BamHI restrictions sites. The resulting plasmids were then transformed into chemically competent E. coli XL-1 Blue cells (Agilent). The entire reading frame of each plasmid was verified by DNA sequencing, after which the plasmids were transformed into chemically competent E. coli BL21-Gold (DE3) cells (Agilent) for protein expression.

**Protein expression and purification.** Proteins for chemical denaturation assays and circular dichroism measurements were expressed using Luria-Bertani (LB) broth supplemented with 100 μg/mL ampicillin. Proteins for NMR spectroscopy were instead expressed using M9 minimal expression medium supplemented with 1 g/L $^{15}$N-ammonium chloride and/or 3 g/L $^{13}$C-D-glucose for isotopic enrichment. Cultures were grown at 37 °C with shaking to an OD600 of ~0.6 after which protein expression was initiated with 1 mM isopropyl β-D-1-thiogalactopyranoside. Following overnight incubation at either 15 °C or 37 °C with shaking (250 rpm) for cultures grown in LB or M9 medium, respectively, cells were harvested by centrifugation and lysed with an EmulsiFlex-B15 cell disruptor (Avestin). Proteins were purified by immobilized metal affinity chromatography according to the manufacturer's protocol (Qiagen), followed by gel filtration in 10 mM sodium phosphate buffer (pH 7.4) using an ENrich SEC 650 size-exclusion chromatography column (BioRad). Purified samples were concentrated using Amicon Ultracel-3K centrifugal filter units (EMD Millipore).

**Analytical size-exclusion chromatography.** Analytical size-exclusion chromatography measurements were performed with a Superdex 200 Increase column (ÄKTA). Protein samples of 1 mL at a concentration of 1 mM were loaded onto the column and eluted with 10 mM sodium phosphate buffer (pH 7.4) at a flow rate of 1 mL/min. Peak UV absorbance was used to determine the elution time of each sample.

**Circular dichroism and thermal denaturation assays.** Circular dichroism measurements were performed with a Jasco J-815 spectrometer using 650-μL aliquots of each Gβ1 sample at a concentration of 40 μM in 10 mM sodium phosphate buffer (pH 7.4) in a 1-mm path-length quartz cuvette (Jasco). For structural characterization of protein folds, circular dichroism spectra were acquired from 185 to 250 nm, sampled every 1 nm at a rate of 10 nm/min. Three scans were acquired and averaged for each sample. For thermal denaturation assays, samples were heated at a rate of 1 °C per minute, and ellipticity at 208 nm was measured every 2 °C. Tm values were determined by fitting a 2-term sigmoid function with baseline correction[37] using nonlinear least-squares regression. Reversibility was confirmed by comparing circular dichroism spectra acquired before and after thermal denaturation experiments from 185 to 250 nm at 25 °C.

**Chemical denaturation assays.** Chemical denaturation assays were performed in triplicate using protein samples at a 1 mg/mL concentration. Protein aliquots of 25 μL in individual wells of UV-Star 96-well plates (Greiner Bio-One) were mixed with 175 μL of 0–5 M guanidium chloride solutions (12 points, evenly spaced) and incubated at room temperature for an hour. Fluorescence emission spectra were measured for each sample from 300 nm to 450 nm (excitation at 295 nm and step size of 2 nm) using an Infinite M1000 plate reader (Tecan). Fluorescence was integrated and converted into fraction of unfolded protein values. Error on these values is reported as standard deviation from three replicates at each denaturant concentration for each Gβ1 variant. Cm values (concentration of denaturant at midpoint of denaturation) were determined by fitting a 2-term sigmoid function using nonlinear least-squares regression.

**NMR spectroscopy.** $^{15}$N- and $^{13}$C-labelled Gβ1 samples for NMR consisted of 0.1–2.0 mM uniformly labelled protein in 10 mM sodium phosphate buffer (pH 7.4), 10 μM EDTA, 0.02% sodium azide, 1× cOmplete EDTA-free Protease Inhibitor Cocktail (Roche), and 10% D₂O for experiments requiring detection of amide protons or 99% D₂O otherwise. All NMR experiments were performed on a Bruker AVANCEIII HD 600 MHz spectrometer equipped with a triple resonance cryoprobe. HSQC, chemical shift assignment, and NOESY experiments were performed at 25 °C, and ZZ-exchange experiments were performed at temperatures varying from 5 °C to 25 °C. NMR data sets were processed with the NMRPipe software package[38] and spectra were analyzed with Topspin v3.5 (Bruker) and NMRViewJ

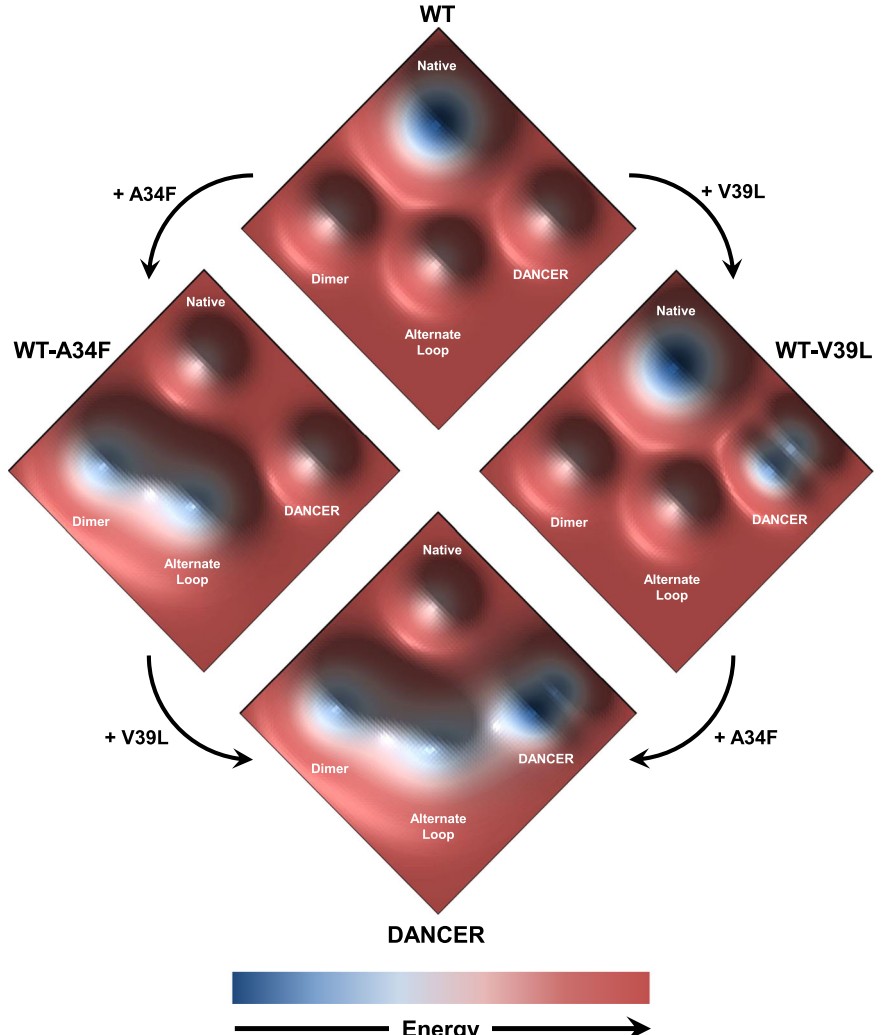

**Fig. 4** Proposed model for the origin of conformational exchange in DANCERs. Major conformational states sampled by Gβ1 variants are shown on a hypothetical energy landscape. Hypothetical energy well depths are based on NMR and molecular dynamics data suggesting which states can be sampled by each variant, and on measured equilibrium constants between these states. Though the DANCER energy well contains two distinct structural states (e.g., core-buried and solvent-exposed states), the energy barrier between these states is much lower than those separating them from either the native or alternate loop states. Therefore, a single DANCER state is indicated for simplicity.

(One Moon Scientific)[39]. Backbone and side-chain chemical shift assignments were obtained from the standard suite of 3D triple resonance experiments, including HSQC, HNCO, HNCACB, and CBCA(CO)NH spectra for backbone assignments, and NH-TOCSY (with mixing time ($\tau_{mix}$ = 75 ms)), CCH-TOCSY ($\tau_{mix}$ = 32.5 ms), and HCCH-TOCSY ($\tau_{mix}$ = 27.5 ms) spectra for side-chain assignments of D3-F34A. Dimer $K_d$ values were calculated using peak volumes for linked major and minor state peaks for Thr17 taken at four different concentrations per protein sample tested. Average backbone amide chemical shift differences ($\Delta\delta$) between W43ε major and minor states were calculated using $\Delta\delta = ((\Delta\delta_{HN})^2 + (\Delta\delta_N/5)^2)^{0.5}$ where $\Delta\delta_{HN}$ and $\Delta\delta_N$ are chemical shift differences calculated for protons and nitrogen atoms, respectively. To obtain D3-F34A distance restraints, $^{15}$N-edited and $^{13}$C-edited HSQC-NOESY ($\tau_{mix}$ = 120 ms) spectra were acquired. ZZ-exchange spectra were acquired at temperatures between 9 and 30 °C for which Trp43ε and Thr17 peaks could be deconvolved from neighboring peaks (or Trp43ε and Thr18 peaks for WT-A34F/V39L and D3-I7L as Thr17 crosspeaks could not be deconvolved at any temperature), and analyzed by fitting a Gaussian model to peaks in distinct clusters and integrating using NMRDraw[38]. Rates of exchange were determined by fitting 4-term relaxation and exchange curves[40] using non-linear least-squares regression, and thermodynamic parameters were determined by fitting exchange rates and temperatures to the Eyring equation.

**Structure determination.** TALOS+[41] was used to determine secondary structure propensities and backbone dihedral restraints for D3-F34A on the basis of measured chemical shifts for $^1$H$_\alpha$, $^{15}$N, $^{13}$C', $^{13}$C$_\alpha$, and $^{13}$C$_\beta$ chemical shifts. Simultaneous NOE assignment and structure calculation was performed using CYANA 2.1[42]. Chemical shifts from cross peaks in 3D $^{15}$N-edited and $^{13}$C-edited NOESY-

HSQC spectra were used as input, in addition to TALOS+ derived dihedral angle restraints. A total of 618 unique and non-redundant distance restraints were used to calculate 100 conformers, and NMR ensembles were represented by the 10 lowest energy conformers. No distance violations >0.5 Å or torsion angle violations >5° were observed in the resulting structures.

**Molecular dynamics simulations.** Structures of DANCER-3, D3-L39I, D3-L39V, and D3-F34A were generated from the NMR structure of D3-L39I by introducing mutations using the TRIAD protein design software (Protabit LLC, Pasadena, CA). Prior to molecular dynamics, the structures were parameterized with the AMBER FF14SB forcefield using the LEaP program from the amber suite (http://ambermd.org/). The protein was surrounded in a cubic box of TIP3P water with 6 Å as the shortest clearance on a single side, and the electrostatic charge neutralized by addition of sodium ions.

Molecular dynamics simulations were performed using in-house code written with the OpenMM API (version 7, http://openmm.org/). A Langevin thermostat was used with a temperature of 298.15 K and a timestep of 2 fs. A Monte Carlo barostat was employed with a pressure of 1 atm, periodic boundary conditions and particle mesh Ewald summation with a long-range cutoff of 12 Å. The lengths of all bonds to hydrogen atoms were constrained. Energy minimization was performed until a convergence of 10 kJ/mol, followed by 100 ns of equilibration. Following equilibration, five production replicas were run for 3 μs each, for a total of 15 μs of simulation time per protein. All analysis of the molecular dynamics trajectories was performed using VMD (www.ks.uiuc.edu/Research/vmd/) on the final 2 μs of each replica trajectory for each protein. Thus, a total of 10 μs simulation time per protein was used for analysis.

**Statistics and reproducibility**. Experiments were repeated in duplicate or triplicate where feasible. All replications were successful and the resulting data is presented with error values representing the standard deviation between replicates. HSQC spectra for all tested proteins were acquired on at least two biological replicates to verify reproducibility. Molecular dynamics simulations were run as five production replicas. No data was excluded from analyses.

**Reporting summary**. Further information on research design is available in the Nature Research Reporting Summary linked to this article.

## Data availability

Structure coordinates for D3-F34A have been deposited in the Protein Data Bank with the accession code 6NJF. NMR data for D3-F34A has been deposited in the Biological Magnetic Resonance Data Bank with the accession code 30532. The datasets generated during and/or analysed during the current study are available from the corresponding author on reasonable request.

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

## Acknowledgements

R.A.C. acknowledges an Early Researcher Award from the Ontario Ministry of Economic Development & Innovation (ER14-10-139), and grants from the Natural Sciences and Engineering Research Council of Canada (RGPIN-2016-04831) and the Canada Foundation for Innovation (26503). N.K.G. acknowledges a grant from NSERC (RGPIN-2011-20298378). A.M.D. is the recipient of a NSERC postgraduate scholarship. A.B. is the recipient of a postdoctoral fellowship from NSERC. M.M.M. is the recipient of an Ontario Graduate Scholarship. We acknowledge Dr. Glenn Facey for assistance with NMR experiments.

## Author contributions

A.M.D. and R.A.C. conceived the project. A.M.D. and M.M.M. performed biophysical characterization experiments. A.B. performed all computational experiments. N.K.G. and A.M.D. designed NMR experiments and analyzed data. A.B. and R.A.C. designed computational experiments. All authors wrote the manuscript.

## Competing interests

The authors declare no competing interests.
