## [Peer Review File · Communications Biology]

Reviewers' comments:

Reviewer #1 (Remarks to the Author):

The central hypothesis in the study of folding is that the tertiary structure is encoded in the primary sequence of a protein. While this remains a largely unresolved puzzle, more and more evidence indicates that the function of a protein is not only governed by its structure but also directly linked to specific conformational dynamics. This presents an even bigger challenge in the study of protein folding and structure-function relationship. Recent progresses in this field have been made by direct evolution-based studies in which novel protein functions were found originated from new dynamic regimes lined to functional sites. In most studies the function-relevant protein dynamics preexisted but was amplified and modulated in response to the change of sequence and structure. The authors recently published a new computational method, termed meta-multistate design, to engineer protein dynamics (Davey, J. A. et al. Rational design of proteins that exchange on functional timescales. *Nature chemical biology* 13, 1280-1285, 2017), in which they created a series of Streptococcal protein G domain β 1 (G β 1) variants, called DANCERS. Through NMR ZZ-exchange measurements and structure determination they demonstrated that these G β 1 variants indeed underwent exchanges between predicted conformational states, as characterized by W43 side-chain orientations, on the millisecond timescale and this specific mode of exchanges cannot be observed in the wild-type protein. The current work is a follow-up of their previous one. They chose DANCER-3, a G β 1 variant containing five-point mutations (Y3F/L7I/A34F/V39L/V54I), as a model system and mutated each individual substitutions back to the wild-type to monitor the effects on the protein dynamics and structure. After series of mutagenesis and NMR studies, they narrowed down that only two mutations, V39L and A34F, are enough to create the previously observed conformational exchange. They concluded that the two mutations work synergistically, i.e. V39L destabilized the native G β 1 structure and A34F allowed the two new conformational states to be accessed on the energy landscape.

The study of protein folding and dynamics is fundamental, fascinating, but very challenging. Although small, G β 1 is a nice model system for such a study. Given the important finding and completeness of the work described in the manuscript, I feel that it deserves a publication in *Communications Biology*, which encourages multidisciplinary studies and contributions from early-career researchers, after the following questions and concerns are fully addressed.

1. In their previous publication (*Nature chemical biology* 13, 1280-1285, 2017), all the G β 1 variants, including DANCER-3, showed similar elution profiles in size-exclusion chromatography as for the wild-type protein, which is monomeric, even at high concentrations of 1-10 mM (Supplementary Figure 4). In this work, however, DANCERS-3 and other variants showed elution profiles similar to that of WT-A34F (Supplementary Figure 6), which is primarily dimeric at concentrations higher than 0.2 mM (*Proteins* 71, 1420-1431, 2008; *J. Phys. Chem. B*, 112, 6008-6012, 2008). Why the results are different?

2. From relaxation-dispersion NMR measurements, at NMR concentration, the WT-A34F mutant showed a slow dimer-monomer exchange process at an average rate ~ 400 (s⁻¹) with a population of $\sim 10\%$ for the monomer (*J. Phys. Chem. B*, 112, 6008-6012, 2008). The extracted average dissociation constant value through NMR concentration titration is $K_d \sim 27$ μ M (*Proteins* 71, 1420-1431, 2008). Similar K_d values for other dimeric G β 1 variants are derived using the same method in this work (Supplementary Table 1) although the value for D3-L39I, also called NERD-S (*Nature chemical biology* 13, 1280-1285, 2017) is smaller, ~ 16 μ M. Now that the K_d value was measured, the monomer W43e resonance should be observable at lower concentrations. Then, what is the chemical shift difference, $\Delta\delta$, between the dimer (major peak) and monomer (minor peak), ~ 0.3 ppm or ~ 1.0 ppm? This spectrum should be shown in either Fig 1 or Supplementary Material because it features whether the system undergoes monomer-dimer exchange only or with additional W43 side-chain

reorientation. This is also important for testing the hypothesis that “only the monomeric form of DANCERS is capable of undergoing Trp43 conformational exchange” (line # 170-171).

3. The result of MD simulations characterizes the dynamic processes on the μ s timescale and thus lacks direct support to the ZZ-exchange NMR result on the ms timescale.

Reviewer #2 (Remarks to the Author):

This manuscript is a rigorous and fascinating dissection of how conformational dynamics can be engineered into a small protein. While the relationship between protein sequence and structure is now (relatively) well understood, the relationship between protein sequence and dynamics--and thus protein sequence and function--remains poorly understood. The authors provide a well worked-out example for how a protein can acquire new conformational exchange with only two mutations. The work is extremely well executed, the problem of evolving dynamics important, and the writing mostly clear. I strongly support publication of this manuscript.

I will first describe the main claims of the manuscript, as I understood them, and then move on to several suggestions and questions.

The authors previously took a protein with a single conformation and then engineered it to exhibit exchange between two conformations. In this manuscript, they did a careful biophysical study to identify how changes in the sequence of the protein led to the new dynamics. They found that only two of the previously identified five mutations were sufficient to convert the original protein into a protein that exchanged between states. They then used a combination of NMR, molecular dynamics simulations, and solution thermodynamics experiments to discover the effects of these mutations. One mutation “pre-stabilized” one of the new conformational states, the other stabilized the second state and the transition state between the conformations. When introduced together, conformational exchange is achieved. The authors speculate that this could be a way for proteins to evolve new dynamics de novo.

Comments, organized in order from most to least significant:

1. The authors should better introduce DANCER-3 for readers not deeply familiar with their previous work. There are a few aspects of this. Most importantly, I do not believe the authors ever explicitly said that WT and DANCER-3 differed in being a monomer vs. monomer/dimer equilibrium. In the first section of the results, they start worrying about dimerization--but I had no idea of why. I pieced together that there was a difference in oligomeric state through the rest of the text and supplement; it would be very helpful to have up front. It was also not explicitly stated that DANCER-3 was one specific DANCER from a set of possibilities. It is implicit in the statement “We chose DANCER-3 for this analysis...” (l. 67), but that was a bit confusing.

I would recommend a short paragraph (cartoon?) that better introduces DANCER-3 vs. WT vs. the set of other DANCERS. A table of sequence differences with which amino acid was in which WT and which was in DANCER-3 (maybe in Fig 1?) would also be helpful.

2. I found the Arrhenius analysis shown in Fig 2 convincing, but it was quite challenging to figure out. This argument would be much easier to understand with a cartoon. (I had to draw one to figure out the argument!). Maybe show Trp43 and Thr17 on a structure at high and low temperatures, with dimerization? It would also be helpful to include the L39I data in the figure--with a note that it was

published previously, obviously--since it is important for interpreting the results.

3. The evolutionary speculation could be set up more effectively. As set up now, the authors posit that V39L first would be a more likely evolutionary pathway. This is difficult to support given that the whole system is designed. An alternate approach that avoids this might be: "We showed that it only takes two mutations to get these dynamics. But that doesn't mean it's a relevant way for how real proteins evolve. If both mutations were really nasty, this new state would be evolutionarily inaccessible. But, amazingly, we found that one mutation doesn't have much detectable effect on the starting state, but sets up the evolution of a new function. This looks like a permissive mutation seen in real evolution (maybe cite Joe Thornton here, who I think was the first one to use this terminology?). As such, we expect that real proteins could evolve this sort of dynamics."

4. These experiments were done using a tiny, biophysical protein. Can these experiments be related to larger, biological proteins? If so, in what way? (I might naively think that this sort of conformational exchange would be difficult to construct in a larger protein with more contacts to "rewire" to shift the energy landscape for the conformational ensemble.)

5. l. 95: "To verify whether ..." I think "verify" should be "distinguish." The authors are telling two possibilities apart, not verifying a conclusion already reached.

Disclosure: I reviewed this manuscript together with a PhD student in my lab.

Reviewer #3 (Remarks to the Author):

This manuscript describes in detail the effects of mutations in a GB1 variant (DANCER-3), and reveals that two mutations are required to induce conformational exchange dynamics. The authors show, using a suite of NMR and biophysical experiments as well as MD simulations, that residues 34 and 39 are crucial for the conformational exchange event that involves outward flip of Trp43.

Briefly, the proposed mechanism underlying the movement of W43 is based on a steric clash (between W43 and F34) and a mutation (V39L) that hampers the protein from adapting its structure to compensate for this steric clash. Consequently, the only possibility to circumvent the steric clash is the outward motion of W43.

The study is well done and besides the somewhat cumbersome naming of mutants (see below) is well written and understandable.

I have only one somewhat minor point that the authors should consider. Due to their readout method of dynamics, namely NMR, the authors have a particular focus on the millisecond time scale, because this is the one that generates peak duplication and EXSY profiles. However, this view is simply biased by the method, and I think it would be helpful for the reader to discuss the possibility that other mutants may as well undergo conformational exchange, but on different time scales.

Furthermore, for better readability, it would be very helpful for the reader if the mutations that distinguish DANCER from WT protein were clearly labeled, e.g. as part of Figure 1, where one can easily find them. It is fairly cumbersome to keep track of the mutations, and to keep in mind what differences e.g. WT-A34F, D3-F34A have.

The text, e.g. page 9, mentions secondary structure names (such as "strand 3"). Please indicate those in the figures, at least in Figure 3.

Response to Reviewers and List of Changes

Reviewer #1:

The central hypothesis in the study of folding is that the tertiary structure is encoded in the primary sequence of a protein. While this remains a largely unresolved puzzle, more and more evidence indicates that the function of a protein is not only governed by its structure but also directly linked to specific conformational dynamics. This presents an even bigger challenge in the study of protein folding and structure-function relationship. Recent progresses in this field have been made by direct evolution-based studies in which novel protein functions were found originated from new dynamic regimes lined to functional sites. In most studies the function-relevant protein dynamics preexisted but was amplified and modulated in response to the change of sequence and structure. The authors recently published a new computational method, termed meta-multistate design, to engineer protein dynamics (Davey, J. A. et al. Rational design of proteins that exchange on functional timescales. *Nature chemical biology* 13, 1280-1285, 2017), in which they created a series of Streptococcal protein G domain β 1 ($G\beta$ 1) variants, called DANCERS. Through NMR ZZ-exchange measurements and structure determination they demonstrated that these $G\beta$ 1 variants indeed underwent exchanges between predicted conformational states, as characterized by W43 side-chain orientations, on the millisecond timescale and this specific mode of exchanges cannot be observed in the wild-type protein. The current work is a follow-up of their previous one. They chose DANCER-3, a G??1 variant containing five-point mutations (Y3F/L7I/A34F/V39L/V54I), as a model system and mutated each individual substitutions back to the wild-type to monitor the effects on the protein dynamics and structure. After series of mutagenesis and NMR studies, they narrowed down that only two mutations, V39L and A34F, are enough to create the previously observed conformational exchange. They concluded that the two mutations work synergistically, i.e. V39L destabilized the native $G\beta$ 1 structure and A34F allowed the two new conformational states to be accessed on the energy landscape.

The study of protein folding and dynamics is fundamental, fascinating, but very challenging. Although small, $G\beta$ 1 is a nice model system for such a study. Given the important finding and completeness of the work described in the manuscript, I feel that it deserves a publication in *Communications Biology*, which encourages multidisciplinary studies and contributions from early-career researchers, after the following questions and concerns are fully addressed.

1. In their previous publication (*Nature chemical biology* 13, 1280-1285, 2017), all the $G\beta$ 1 variants, including DANCER-3, showed similar elution profiles in size-exclusion chromatography as for the wild-type protein, which is monomeric, even at high concentrations of 1-10 mM (Supplementary Figure 4). In this work, however, DANCERS-3 and other variants showed elution profiles similar to that of WT-A34F (Supplementary Figure 6), which is primarily dimeric at concentrations higher than 0.2 mM (*Proteins* 71, 1420-1431, 2008; *J. Phys. Chem. B*, 112, 6008-6012, 2008). Why the results are different?

Response:

Reviewer #1 is correct in that we previously obtained similar elution profiles for DANCER-3 and wild-type $G\beta$ 1 during size-exclusion chromatography (SEC), while in this present study, we observed a significant difference. We attribute this discrepancy to the resolution of the SEC protocol that we previously reported, which involved the use of a different FPLC instrument and column. Specifically, we previously used a BioLogic DuoFlow system equipped with an ENrich SEC 650 column (both from Bio-Rad). According to the manufacturer's specifications, the resolution of the Enrich SEC 650 column should have

been sufficient for separating G β 1 monomers from dimers, assuming no monomer/dimer exchange. However, we discovered after publication of our *Nature Chemical Biology* article that the Enrich SEC 650 column cannot in practice separate monomeric G β 1s from weak exchanging dimers, such as DANCER variants containing the A34F mutation. In contrast, we were able to resolve monomers from the mixed monomer-dimer populations using a similar SEC protocol (i.e., similar protein concentrations and identical buffer conditions) with the ÄKTA Pure system equipped with a Superdex 200 Increase column (GE Healthcare), as described in this current study.

We have explained the discrepancy with our previous results in the legend to Supplementary Figure 6:

“Supplementary Figure 6. Analytical size-exclusion chromatograms of G β 1 variants. Chromatograms monitored by absorption at 280 nm using an ÄKTA Pure system equipped with a Superdex 200 Increase 10/300 column (GE Healthcare) show a shift of approximately 1 mL in elution volume for dimeric G β 1 variants relative to the monomeric wild type (WT), shown in grey for comparison. These variations in elution volume between monomeric and dimeric variants were not observed previously using similar elution conditions on a Bio-Rad BioLogic DuoFlow system equipped with an ENrich SEC 650 column, likely reflecting the difference in resolving power of the two systems. (Ref: 16)”

2. From relaxation-dispersion NMR measurements, at NMR concentration, the WT-A34F mutant showed a slow dimer-monomer exchange process at an average rate ~ 400 (s $^{-1}$) with a population of $\sim 10\%$ for the monomer (J. Phys. Chem. B, 112, 6008-6012, 2008). The extracted average dissociation constant value through NMR concentration titration is $K_d \sim 27$ μ M (Proteins 71, 1420-1431, 2008). Similar K_d values for other dimeric G β 1 variants are derived using the same method in this work (Supplementary Table 1) although the value for D3-L39I, also called NERD-S (Nature chemical biology 13, 1280-1285, 2017) is smaller, ~ 16 μ M. Now that the K_d value was measured, the monomer W43e resonance should be observable at lower concentrations. Then, what is the chemical shift difference, $\Delta\delta$, between the dimer (major peak) and monomer (minor peak), ~ 0.3 ppm or ~ 1.0 ppm? This spectrum should be shown in either Fig 1 or Supplementary Material because it features whether the system undergoes monomer-dimer exchange only or with additional W43 side-chain reorientation. This is also important for testing the hypothesis that “only the monomeric form of DANCERs is capable of undergoing Trp43 conformational exchange” (line # 170-171).

Response:

We did measure the ^1H - ^{15}N HSQC spectrum of D3-L39I (NERD-S) at a lower concentration (20 μ M) and observed an additional set of peaks, including a low-intensity W43e peak with a chemical shift difference of ~ 0.8 ppm with its corresponding major peak (see spectrum below). However, we were unable to accurately quantify ZZ-exchange kinetics for this set of peaks given the low signal-to-noise ratio. These minor state peaks were not detectable at concentrations strongly favoring the dimer (0.1 – 1.0 mM), helping to confirm that this exchange occurs only in the monomer.

Supplementary Figure 10. D3-L39I (NERD-S) displays evidence of W43 conformational exchange at a concentration of 20 μM . The ^1H - ^{15}N HSQC spectrum of D3-L39I at 20 μM is shown on the left ($K_d = 16 \pm 2 \mu\text{M}$). The inset shows the W43 ϵ minor state peak identified using ZZ-exchange experiments linked to its corresponding major state peak with a double-sided arrow. The chemical-shift displacement ($\Delta\delta$) between major and minor state peaks is indicated.

To address this comment, we have added an additional supplementary figure showing the low-concentration HSQC spectrum of D3-L39I (see above), as well as the following text in the results section:

“Our prior results with the L39I mutant of DANCER-3 (previously reported as NERD-S, here renamed D3-L39I) corroborate this hypothesis, as D3-L39I exhibits no detectable Trp43 conformational exchange at the concentration range used in our NMR experiments (0.1 – 1 mM) despite significant sequence and spectral similarity to DANCER-3. At a concentration of 100 μM , the monomeric form of DANCER-3 was observable while that of D3-L39I was not. However, when the concentration of D3-L39I was lowered to 20 μM , which is close to its K_d (Supplementary Table 1), we observed the appearance of a second Trp43 ϵ peak with a $\Delta\delta$ characteristic of Trp43 conformational exchange (Supplementary Figure 10).”

3. The result of MD simulations characterizes the dynamic processes on the us timescale and thus lacks direct support to the ZZ-exchange NMR result on the ms timescale.

Response:

We agree with Reviewer #1 that the timescale of our MD simulations (microseconds) is shorter than that of the ZZ-Exchange experiments (milliseconds). However, comparisons can still be drawn between the two since motions observed at the microsecond timescale

in MD may form part of the millisecond timescale processes revealed by NMR. Specifically, the frequent rotameric interconversions seen in MD at the microsecond timescale for DANCERs (Figure 3 and Supplementary Figure 14) may represent the first step in a process that eventually leads to longer timescale transitions. This may be the case if, for instance, longer timescale transitions require concerted side-chain and backbone movements, whereas the MD simulations capture one or the other but are not long enough to capture their rarer co-occurrence. Thus, differences at the microsecond timescale in MD are still relevant to our comparison of dynamics between GB1 variants.

We have added the following sentence to explain our rationale for using microsecond-timescale MD simulations:

“To gain insights on the role of the conservative V39L mutation in Trp43 conformational exchange and evaluate its underlying synergistic effect with A34F, we performed microsecond-timescale molecular dynamics (MD) simulations on DANCER-3, D3-F34A, and D3-L39V. We also included D3-L39I in this analysis because it contains an isosteric mutation at position 39 that might further help to demonstrate the role of this residue in DANCER dynamics. Although the timescale of these MD simulations is shorter than that of the ZZ-exchange experiments, motions observed at the microsecond timescale in MD may form part of the millisecond timescale processes revealed by NMR.”

Reviewer #2:

This manuscript is a rigorous and fascinating dissection of how conformational dynamics can be engineered into a small protein. While the relationship between protein sequence and structure is now (relatively) well understood, the relationship between protein sequence and dynamics--and thus protein sequence and function--remains poorly understood. The authors provide a well worked-out example for how a protein can acquire new conformational exchange with only two mutations. The work is extremely well executed, the problem of evolving dynamics important, and the writing mostly clear. I strongly support publication of this manuscript.

I will first describe the main claims of the manuscript, as I understood them, and then move on to several suggestions and questions.

The authors previously took a protein with a single conformation and then engineered it to exhibit exchange between two conformations. In this manuscript, they did a careful biophysical study to identify how changes in the sequence of the protein led to the new dynamics. They found that only two of the previously identified five mutations were sufficient to convert the original protein into a protein that exchanged between states. They then used a combination of NMR, molecular dynamics simulations, and solution thermodynamics experiments to discover the effects of these mutations. One mutation “pre-stabilized” one of the new conformational states, the other stabilized the second state and the transition state between the conformations. When introduced together, conformational exchange is achieved. The authors speculate that this could be a way for proteins to evolve new dynamics de novo.

Comments, organized in order from most to least significant:

1. The authors should better introduce DANCER-3 for readers not deeply familiar with their previous work. There are a few aspects of this. Most importantly, I do not believe the authors ever explicitly said that WT and DANCER-3 differed in being a monomer vs. monomer/dimer equilibrium. In the first section of the results, they start worrying about dimerization--but I had no

idea of why. I pieced together that there was a difference in oligomeric state through the rest of the text and supplement; it would be very helpful to have up front. It was also not explicitly stated that DANCER-3 was one specific DANCER from a set of possibilities. It is implicit in the statement “We chose DANCER-3 for this analysis” (l. 67), but that was a bit confusing.

I would recommend a short paragraph (cartoon?) that better introduces DANCER-3 vs. WT vs. the set of other DANCERS. A table of sequence differences with which amino acid was in which WT and which was in DANCER-3 (maybe in Fig 1?) would also be helpful.

Response:

We apologize for not being sufficiently clear in describing our previous work with DANCERS and explaining differences in oligomeric state between the GB1 variants. To clarify, we have added details on DANCERS to the introduction (changes highlighted in yellow):

“Previously, we used computational design to create three Streptococcal protein G domain β 1 ($G\beta$ 1) variants termed DANCERS (DANCER-1, DANCER-2, and DANCER-3) that undergo a specific mode of conformational exchange that has never been observed in the wild-type protein. This exchange involves spontaneous switching of the Trp43 side chain between core-buried and solvent-exposed states, which is accompanied by rotameric changes to residues in the protein core and by displacement of the helix relative to the β -sheet that nevertheless preserves the native $G\beta$ 1 fold (Fig. 1a). The successful generation of DANCERS involved simultaneous introduction of 5 or 6 mutations to the $G\beta$ 1 core that were predicted to stabilize both target states and create a sufficiently small transition state barrier to allow interconversion on a functional timescale.”

We have also modified Table 1 (see below) to highlight more clearly the mutations found in each variant studied herein (including DANCER-3), and edited the first paragraph of the “Results” section to better justify our selection of DANCER-3 as case study and to better introduce the question of the dimer:

“To investigate how sequence changes can lead to novel modes of conformational exchange in a globular protein, we individually reverted each of the five mutations present in DANCER-3 to the corresponding amino acid found in WT $G\beta$ 1 (Fig. 1a,b). We chose DANCER-3 for this analysis over other DANCERS as its slow exchange rate and distinct 1H and ^{15}N chemical shifts for both Trp43 conformational states facilitate analysis of exchange using nuclear magnetic resonance (NMR) spectroscopy. [...] Given the difference in NMR spectra between D3-F34A and all other DANCER mutants studied here, we investigated the possibility that DANCER variants containing a Phe residue at position 34 (Table 1) form dimers in solution, as the A34F mutation has been shown to induce dimerization when introduced on its own into WT $G\beta$ 1. (Ref: 25,31) Analytical size-exclusion chromatography using WT $G\beta$ 1 and its A34F point mutant (WT-A34F) as controls for monomeric and dimeric proteins, respectively, revealed that unlike WT $G\beta$ 1, which exists as a monomer under these conditions, DANCER-3 and all its variants containing the A34F mutation form weak dimers with K_d values measured by NMR between 31 and 240 μ M (Supplementary Fig. 6 and Supplementary Table 1).”

Table 1. Stability of Gβ1 variants

Protein	Mutations from Wild Type (WT)					T _m ^a (°C)	C _m ^b (M)	ΔG _U ^c (kcal/mol)
	Y3	L7	A34	V39	V54			
WT	-	-	-	-	-	81	2.66 ± 0.04	4.3 ± 0.3
WT-A34F	-	-	F	-	-	57	1.02 ± 0.04	2.7 ± 0.2
WT-V39L	-	-	-	L	-	74	2.32 ± 0.02	4.3 ± 0.4
WT-A34F/V39L	-	-	F	L	-	61	1.34 ± 0.01	3.0 ± 0.1
DANCER-3	F	I	F	L	I	71	1.72 ± 0.05	3.4 ± 0.4
D3-F3Y	-	I	F	L	I	66	1.66 ± 0.06	3.3 ± 0.3
D3-I7L	F	-	F	L	I	60	1.30 ± 0.03	2.8 ± 0.4
D3-F34A	F	I	-	L	I	84	2.74 ± 0.07	4.3 ± 0.1
D3-L39V	F	I	F	-	I	68	1.75 ± 0.05	2.9 ± 0.1
D3-I54V	F	I	F	L	-	65	1.82 ± 0.06	3.2 ± 0.4

^a Thermal denaturation midpoint temperature determined through loss of circular dichroism signal at 208 nm

^b Concentration of guanidinium chloride at denaturation midpoint (25 °C, n = 3, mean ± s.d.)

^c Free energy of unfolding determined by chemical denaturation with guanidinium chloride at 25 °C (n = 3, mean ± s.d.)

Lastly, we have modified Figure 1 and its legend to emphasize more clearly the mutations found in DANCER-3 relative to WT GB1 (Fig. 1a and 1b).

2. I found the Arrhenius analysis shown in Fig 2 convincing, but it was quite challenging to figure out. This argument would be much easier to understand with a cartoon. (I had to draw one to figure out the argument!). Maybe show Trp43 and Thr17 on a structure at high and low temperatures, with dimerization? It would also be helpful to include the L39I data in the figure--with a note that it was published previously, obviously--since it is important for interpreting the results.

Response:

We apologize for not being sufficiently clear in our description of Figure 2, and acknowledge as pointed out by Reviewer #2 that the interpretation of these results relies on a good understanding of the thermodynamics of the system. To assist in this interpretation, we have added a cartoon representation of DANCER exchange kinetics at high and low temperature to Figure 2, and have modified accordingly the figure legend.

“c. In DANCERs, non-Arrhenius behavior for W43ε transitions is observed due to an altered monomer-dimer equilibrium. As dimerization of DANCERs is favored at high temperature and W43 exchange is inhibited in the dimer, observed rates (k_{obs}) of W43 exchange are reduced when temperature increases. In contrast, k_{obs} for exchange of the T17 residue found at the dimer interface increase with temperature, indicating that dimerization kinetics follow Arrhenius behavior. Non-Arrhenius behavior for W43ε transitions was not detected in Gβ1 variants that do not undergo W43 conformational exchange, suggesting that all kinetics measured for these variants report only on dimerization.”

Regarding the ZZ-exchange data for D3-L39I, it is unfortunately too noisy for us to be able to calculate accurate exchange rates. Therefore, we do not feel comfortable including these data in our manuscript.

3. The evolutionary speculation could be set up more effectively. As set up now, the authors posit that V39L first would be a more likely evolutionary pathway. This is difficult to support given that the whole system is designed. An alternate approach that avoids this might be: “We showed that it only takes two mutations to get these dynamics”. But that doesn’t mean it’s a relevant way for how real proteins evolve. If both mutations were really nasty, this new state would be evolutionarily inaccessible. But, amazingly, we found that one mutation doesn’t have much detectable effect on the starting state, but sets up the evolution of a new function. This looks like a permissive mutation seen in real evolution (maybe cite Joe Thornton here, who I think was the first one to use this terminology?). As such, we expect that real proteins could evolve this sort of dynamics.

Response:

To address this comment, we have reworded the second paragraph of the discussion as follows:

“Epistasis caused by ruggedness in the protein energy landscape complicates analysis of the evolution of new modes of conformational exchange in natural proteins. However, the G β 1 variants described here provide a unique opportunity to start addressing this question. Although the conformational exchange of Trp43 observed in DANCERs was designed computationally, we showed herein that only two mutations are necessary for this dynamic exchange to arise. There are thus only two possible evolutionary pathways by which these mutations might be introduced (Fig. 4). As protein stability is an important determinant of evolutionary fitness (Ref: 32), if both mutations were highly destabilizing, DANCERs would be evolutionarily inaccessible. However, the V39L mutation has no significant effect on protein structure, stability, and function in the context of the WT G β 1 sequence. Thus, it strongly resembles the “permissive mutations” observed during protein evolution (Ref: 33), which have no immediate structural or functional effect but stabilize specific structural elements such that the protein can tolerate ensuing mutations causing a functional change. In the context of DANCERs, we postulate that V39L is a permissive mutation that reshapes the protein’s energy landscape to disfavor the alternate loop conformation by stabilizing the helix at its C-terminus and thereby favor Trp43 conformational exchange without significantly altering the energy well depth of the native fold. Conversely, the A34F mutation is highly destabilizing to the native fold and promotes the alternate loop conformation, facilitating access to other energy wells in the surrounding landscape. This explains the synergistic effects of these two mutations, where one destabilizes the native energy well and the other makes neighboring wells more accessible in a way that allows novel dynamic motions to develop.”

4. These experiments were done using a tiny, biophysical protein. Can these experiments be related to larger, biological proteins? If so, in what way? (I might naively think that this sort of conformational exchange would be difficult to construct in a larger protein with more contacts to “rewire” to shift the energy landscape for the conformational ensemble.)

Response:

Although larger proteins are more complex systems for studying and designing dynamics, they are not necessarily more stable than small proteins. Indeed, mutations that cause free energy changes of a few kcal/mol have been shown to have a drastic effect on protein stability, conformation, and equilibrium when introduced into proteins larger than GB1. For example, Fraser and colleagues (Fraser *Nature* 2009, Otten *Nature Communications* 2018) demonstrated that conformational exchange of a phenylalanine side-chain in the

proline isomerase CypA, an 18 kDa protein that catalyzes a metabolic reaction, could be abolished with a single mutation, and recovered with two further mutations. This example demonstrates the applicability of our experiments and results to larger proteins.

To address these comments, we have reworded the third paragraph of the discussion as follows:

“[...] This process parallels previously studied evolutionary processes wherein the development of new dynamics-linked function occurs via the introduction of mutations that amplify dynamics, followed by selection against undesirable competing motions that negatively impact activity or stability. (Ref: 10) Previous studies have demonstrated how a small number of mutations can abolish and re-establish conformational exchange pathways in larger natural proteins (Ref: 3, 35), as we observed with our model system comprised of a relatively small protein. These trends support the general idea that permissive mutations in globular proteins can provide a route for the introduction of new dynamic modes in the molecular evolution of dynamics-linked functions, shedding light on the link between protein sequence and dynamics.

5. I. 95: “To verify whether” I think “verify” should be “distinguish”. The authors are telling two possibilities apart, not verifying a conclusion already reached.

Response:

We have changed this line according to Reviewer #3’s suggestion.

Disclosure: I reviewed this manuscript together with a PhD student in my lab.

Reviewer #3

This manuscript describes in detail the effects of mutations in a GB1 variant (DANCER-3), and reveals that two mutations are required to induce conformational exchange dynamics. The authors show, using a suite of NMR and biophysical experiments as well as MD simulations, that residues 34 and 39 are crucial for the conformational exchange event that involves outward flip of Trp43. Briefly, the proposed mechanism underlying the movement of W43 is based on a steric clash (between W43 and F34) and a mutation (V39L) that hampers the protein from adapting its structure to compensate for this steric clash. Consequently, the only possibility to circumvent the steric clash is the outward motion of W43. The study is well done and besides the somewhat cumbersome naming of mutants (see below) is well written and understandable.

I have only one somewhat minor point that the authors should consider. Due to their readout method of dynamics, namely NMR, the authors have a particular focus on the millisecond time scale, because this is the one that generates peak duplication and EXSY profiles. However, this view is simply biased by the method, and I think it would be helpful for the reader to discuss the possibility that other mutants may as well undergo conformational exchange, but on different time scales.

Response:

We agree that new modes of dynamics may exist in mutants we have classified as non-exchangers on timescales that we did not investigate (e.g. ps – ns, ns – μ s). However, we

chose to focus on dynamics on the timescale that was targeted in computational design, (i.e. the millisecond timescale). Dynamics on other timescales may underlie the designed conformational exchange, as alluded to by our MD results, however it was beyond the scope of our study to perform a detailed characterization of dynamics at all possible timescales. One of the complications of characterizing faster timescale motions is that it is difficult to obtain structural insight into these motions, with amplitudes and timescales being obtained with no direct structural data. Even if faster timescale motions are detected in both DANCERs and non-exchanging mutants, it is not possible to know if these are related to the designed dynamics. In contrast, detailed structural information on slowly exchanging states can be captured in the NOE record, as was done for the DANCER proteins, and shown to be absent in D3-L39I (see Reference 16). It is unlikely that understanding of the origin of millisecond timescale dynamics would have been significantly enhanced by the addition of dynamics data at other timescales, and consequently, we did not pursue the characterization of dynamics at other timescales.

We have added the following statement to address this comment: “Although dynamics may differ between variants at multiple timescales, we focused on those at the millisecond timescale, since these were the target of our original design” (line 110).

Furthermore, for better readability, it would be very helpful for the reader if the mutations that distinguish DANCER from WT protein were clearly labeled, e.g. as part of Figure 1, where one can easily find them. It is fairly cumbersome to keep track of the mutations, and to keep in mind what differences e.g. WT-A34F, D3-F34A have. The text, e.g. page 9, mentions secondary structure names (such as “strand 3”). Please indicate those in the figures, at least in Figure 3.

Response:

We thank Reviewer 3 for the suggestion. We have labelled mutations that distinguish DANCER-3 from WT on the structures of DANCER-3 and WT (Figure 1a and 1b). We have also added labels identifying strand β 3 on the structures in Figure 3, and modified the figure legend accordingly (“[...] which connects the α -helix to strand β 3 (labeled at its N-terminal extremity), [...]”). Lastly, we have updated Table 1 to clearly indicate the mutations found in each variant (see response to Reviewer #2, comment #1).